# *12-Oxophytodienoate Reductase* Overexpression Compromises Tolerance to *Botrytis cinerea* in Hexaploid and Tetraploid Wheat

**DOI:** 10.3390/plants12102050

**Published:** 2023-05-22

**Authors:** Evgeny Degtyaryov, Alexey Pigolev, Dmitry Miroshnichenko, Andrej Frolov, Adi Ti Basnet, Daria Gorbach, Tatiana Leonova, Alexander S. Pushin, Valeriya Alekseeva, Sergey Dolgov, Tatyana Savchenko

**Affiliations:** 1Institute of Basic Biological Problems, Pushchino Scientific Center for Biological Research, Russian Academy of Sciences, 142290 Pushchino, Russia; evkras99@yandex.ru (E.D.); alexey-pigolev@rambler.ru (A.P.); miroshnichenko@bibch.ru (D.M.); 2Branch of Shemyakin and Ovchinnikov Institute of Bioorganic Chemistry, Russian Academy of Sciences, 142290 Pushchino, Russia; aspushin@gmail.com (A.S.P.); lera@bibch.ru (V.A.); dolgov@bibch.ru (S.D.); 3Department of Bioorganic Chemistry, Leibniz Institute of Plant Biochemistry, 06120 Halle (Saale), Germany; andrej.frolov@ipb-halle.de (A.F.); aditibasnet66@gmail.com (A.T.B.); daria.gorba4@yandex.ru (D.G.); tatiana.leonova@ipb-halle.de (T.L.); 4Laboratory of Analytical Biochemistry and Biotechnology, Timiryazev Institute of Plant Physiology, Russian Academy of Sciences, 127276 Moscow, Russia

**Keywords:** 12-oxophytodienoate reductase, jasmonates, *Botrytis cinerea*, tolerance, bread wheat, emmer wheat

## Abstract

12-Oxophytodienoate reductase is the enzyme involved in the biosynthesis of phytohormone jasmonates, which are considered to be the major regulators of plant tolerance to biotic challenges, especially necrotrophic pathogens. However, we observe compromised tolerance to the necrotrophic fungal pathogen *Botrytis cinerea* in transgenic hexaploid bread wheat and tetraploid emmer wheat plants overexpressing *12-OXOPHYTODIENOATE REDUCTASE-3* gene from *Arabidopsis thaliana*, while in Arabidopsis plants themselves, endogenously produced and exogenously applied jasmonates exert a strong protective effect against *B. cinerea*. Exogenous application of methyl jasmonate on hexaploid and tetraploid wheat leaves suppresses tolerance to *B. cinerea* and induces the formation of chlorotic damages. Exogenous treatment with methyl jasmonate in concentrations of 100 µM and higher causes leaf yellowing even in the absence of the pathogen, in agreement with findings on the role of jasmonates in the regulation of leaf senescence. Thereby, the present study demonstrates the negative role of the jasmonate system in hexaploid and tetraploid wheat tolerance to *B. cinerea* and reveals previously unknown jasmonate-mediated responses.

## 1. Introduction

Fungal diseases of wheat, causing a significant yield reduction and deterioration of grain quality, pose a serious risk to agriculture [1]. The regulation of plant tolerance to biotic challenges, especially necrotrophic pathogens, is primarily attributed to the phytohormone jasmonates [2,3]. The role of jasmonates in the improvement of plant tolerance to necrotrophic fungi, such as *Fusarium oxysporum*, *Alternaria brassicicola*, *Plectosphaerella cucumerina*, and *Botrytis cinerea*, has been shown in many studies [4,5,6,7,8,9,10,11]. However, there is also ample evidence indicating that the landscape of regulatory functions of jasmonates in plant tolerance to pathogens is more complex and that jasmonate-induced effects vary widely depending on plant and pathogen species, environmental conditions, and the physiological state of the host plant. Thus, according to individual reports, jasmonates can also protect plants against biotrophic pathogens and can even induce host plant sensitivity to necrotrophic pathogens [9,12]. 

Genetic manipulations leading to the alteration of the endogenous level of the hormone in plant tissues represent the most reliable way of elucidating its function. In-depth knowledge of the jasmonate biosynthesis pathway is an important prerequisite for such genetic manipulations. The biosynthesis pathway leading to the formation of jasmonic acid from the fatty acid substrate has been studied in detail, as has the jasmonate-associated signal transduction pathway [2]. The biosynthesis is initiated in chloroplasts, where lipoxygenases oxidize α-linolenic acid to form 13(S)-hydroperoxy-9,11,15-octadecatrienoic acid (13-HPOT) [2,13]. 13-HPOT is further converted to 12-oxo-phytodienoic acid (12-OPDA) by the sequential action of allene oxide synthase and allene oxide cyclase [13,14,15,16,17], and then 12-OPDA is transported to peroxisomes [18,19]. In peroxisomes, the double bond of the cyclopentenone ring of 12-OPDA is reduced by oxophytodienoate reductase (OPR) [20,21], and three cycles of β-oxidation of the carboxylic acid side chain occur, resulting in the formation of jasmonic acid (JA) [22,23,24]. An alternative pathway for JA biosynthesis from hexadecatrienoic acid has also been described [25]. JA can be further modified in the cytoplasm, including the formation of a volatile methyl jasmonate derivative (MeJA) and a conjugate with the amino acid isoleucine (JA-Ile), which is responsible for the regulation of the most JA-dependent processes [26]. The jasmonate signal transduction pathway, involving the intracellular receptor CORONATINE INSENSITIVE 1 (COI1), which is a component of the Skp1-Cullin-F-box ubiquitin ligase complex (SCFCOI1), and the repressor Jasmonate ZIM-domain proteins (JAZ), is also well understood [27,28].

Here we conducted an analysis of transgenic hexaploid bread wheat and tetraploid emmer wheat plants overexpressing the *12-OXOPHYTODIENOATE REDUCTASE 3* gene from *Arabidopsis thaliana* (*AtOPR3*) to elucidate the role of jasmonates in wheat tolerance to necrotrophic pathogens. Emmer wheat *(T. dicoccum* (Schrank., 2n = 4x = 28)) is considered to play a central role in the domestication of wheat and is a direct progenitor of modern cultivated varieties, including the tetraploid durum wheat (*T. durum* Desf., 2n = 4x = 28) and hexaploid bread wheat (*T*. *aestivum* L., 2n = 6x = 42) [29]. Centuries of domestication and subsequent breeding have reduced the genetic diversity of modern wheat cultivars compared to their wild ancestors. The gene pool of emmer wheat appeared earlier and is thought to combine the genetic diversity of wheat’s wild ancestors. In this regard, the study and comparison of jasmonate profiles in various polyploid wheat species could be helpful for expanding knowledge of the regulatory functions of jasmonates in cereals and may potentially contribute to improving resistance against various stresses in modern cultivated wheat.

## 2. Results

### 2.1. Generation of Transgenic Tetraploid Emmer Wheat Plants Overexpressing AtOPR3

In order to study the role of jasmonates in wheat tolerance to necrotrophic pathogens, in addition to previously established transgenic hexaploid wheat plants (*T. aestivum* L. cv. “Saratovskaya-60”) overexpressing the *12-OXOPHYTODIENOATE REDUCTASE 3* gene from *A. thaliana* (*AtOPR3*) [30], we have generated new transgenic cultivated emmer wheat *(T. dicoccum* (Schrank), cv. “Runo”) overexpressing *AtOPR3*. The previously described pBAR-GFP.UbiOPR3 vector, enabling the expression of the *AtOPR3* gene under the control of the *ZmUbi1* constitutive promoter [30], has been used to perform the genetic transformation of the tetraploid “Runo” (Appendix A). A total of 35 independent putative transgenic events were generated following the biolistic delivery of pBAR-GFP. UbiOPR3 vector into 236 morphogenic explants and subsequent dual selection (green fluorescent protein (GFP) fluorescence/herbicide resistance of selected tissues). Primary plants corresponding to 34 events were successfully established in the greenhouse, and 31 of them showed GFP fluorescence in pollen at the heading stage. GFP activity was consistent with the gene expression data generated by the end-point RT-PCR (Figure 1a). *AtOPR3* transcripts were present in the leaves of 26 primary T0 plants as an amplification of a specific 295 bp band, while they were not detected in non-transformed “Runo” or in several GFP-positive transgenic plants (Figure 1a).

Primary T0 transgenic emmer wheat plants overexpressing the introduced *AtOPR3* were normal and fertile. T1 seeds produced after self-pollination of 15 primary T0 plants were further analyzed for heritability of transgenes, and 47% of transformants showed a Mendelian 3:1 segregation ratio according to the chi-square test (Appendix A). Ultimately, homozygous T2 plants of three *AtOPR3*-positive T0 events, designated as RC12, RC26, and RC29, showing the segregation of transgenes as a single functional locus, were selected for further analyses. T4 homozygous families of RC12, RC26, and RC29 were further analyzed by real-time RT-PCR to examine the expression of *AtOPR3* over generations. T4 transgenic plants showed an abundance of *AtOPR3* transcript in leaves, especially the RC-29 line, while non-transgenic parent plants did not accumulate *AtOPR3* mRNA (Figure 1b). 

### 2.2. Analysis of Jasmonate Content in Wheat Plants

To determine whether the overexpression of the *AtOPR3* gene leads to an increase in jasmonate levels in wheat species, we have employed the tetraploid emmer wheat plants described above and previously generated transgenic hexaploid bread wheat plants cv. “Saratovskaya-60” (further designated as Sar-60) [30]. For hormone analysis, all transgenic lines and non-transgenic control plants were grown in greenhouses simultaneously under the same temperature and light conditions, and the wounding treatment was performed as uniformly as possible. We have measured the levels of 12-OPDA, JA, and JA-Ile in transgenic plants and non-transgenic parent lines in intact and wounded leaves. In the first place, we have compared the jasmonate profiles of studied non-transgenic wheat plants. The results of the analysis are presented in Figure 2. A comparison of jasmonate contents between the non-transgenic plants of tetraploid and hexaploid varieties revealed several differences. As expected, the basal levels of jasmonates were very low in tetraploid and hexaploid wheat plants. Thus, the 12-OPDA level ranged from 4.3 to 18.9 pmol/g fresh weight (f.w.) and did not show any difference between Sar-60 and Runo, whereas the JA levels ranged from 5 to 18.5 pmol/g f.w. and were significantly higher in emmer wheat (*p* = 0.0111). The basal JA-Ile levels were the lowest among the analyzed jasmonates in both genotypes, ranging from 2.2 to 8.3 pmol/g f.w. Wounding was shown to promote an increase in the levels of all quantified jasmonates in all transgenic plants, but the magnitude of the changes in the hormone contents differed significantly. Thus, wounding induced a more pronounced increase in the jasmonate contents in Sar-60: the average JA levels were increased about 8-fold in bread wheat, while in emmer wheat this alteration accounted for only 2.6-fold; the 12-OPDA levels were 3.5-fold increased upon wounding in hexaploid wheat and only 1.7-fold in tetraploid wheat. The highest wounding-induced increase was observed for JA-Ile in both genotypes—13.5-fold in Sar-60 and 6.7-fold in Runo.

It is important to note that the Sar-60 plants demonstrated higher intra-group dispersion in comparison to the Runo plants. This observation was confirmed in several independent experiments. This might indicate that the stress-induced changes in jasmonate levels were very dynamic in hexaploid wheat and were affected by multiple internal and external factors. As a result of such variations and due to the low baseline levels of the hormone, it was impossible to assess statistically significant differences in the contents of jasmonates in the leaves of non-transgenic and transgenic hexaploid wheat plants (Figure 3a,c,e). On the other hand, the average (and median) JA and JA-Ile levels were notably higher in transgenic lines SC15 and SC20 in comparison to Sar-60. Remarkably, the tetraploid Runo plants responded to wounding more uniformly. Therefore, we succeeded in identifying transgenic lines with increased stress-induced jasmonate levels (Figure 3b,d,f). The wounded leaves of the transgenic tetraploid lines RC12 and RC29 demonstrated significantly higher JA and JA-Ile levels in comparison to the wounded leaves of non-transgenic Runo.

### 2.3. The Phenomenology of B. cinerea Infection Development on the Detached Wheat Leaves

The use of a detached leaf infection protocol ensures the maintenance of the moisture level necessary for the infection and enables easy monitoring of the symptom development. Before comparing the tolerance of different genotypes to the pathogenic fungi, we studied the development of the infection on individual detached leaves of non-transgenic plants. The development of the *B. cinerea* infection resulted in the formation of a brownish necrotic lesion zone around the inoculation site and a yellow chlorotic zone (Figure 4a). 

In most cases, the boundary between the necrotic and chlorotic zones was clearly delineated, while the boundary between the chlorotic zone and the adjacent green tissues, devoid of visible infection symptoms on the distal parts of the detached leaf fragment, was blurred. To obtain more information about the nature of the observed symptoms, we have stained the infected leaves with trypan blue, which stains dead cells, cells with an increased membrane permeability, while cells with intact membrane integrity remain unstained [31], and also living fungal hyphae [32]. The tissues of the necrotic zone were stained with trypan blue, while the tissues of the chlorotic area remained unstained (Figure 4b). As seen in Figure 4b, inside the necrotic zone, bright blue coloration is present closer to the site of infection, and areas of the necrotic zone more distant from the site of infection are colored very light blue. Larger and denser stained islets are seen closer to the zone of infection, and smaller, more sparse spots are seen closer to the borders with the chlorotic zone, but such spots are not detected outside the necrotic zone (Figure 4c). On leaves with more advanced infection, almost the entire surface of the necrotic zone is stained with trypan blue, although light uncolored spots are still present (Figure 4d). Light microscopy of trypan blue-stained tissue allowed visualization of fungus hyphae and developed mycelia in the necrotic lesion region (Figure 4e), but not in the chlorotic region. The images taken by a digital dissecting microscope in transmitted light mode clearly show the formation of a transparent depigmented area around the spore inoculation site (Figure 4g,h). As the infection progresses, this zone expands, while delineated boundaries distinguishable on macroscopic images are maintained (Figure 4i,j). Over time, the sporulation of fungi is initiated in these damaged areas, and conidiophores begin to appear on the leaf surface and form conidia (Figure 4i–k).

### 2.4. Analysis of Transgenic Lines Tolerance to B. cinerea

Four independent transgenic lines of hexaploid bread wheat (SC15, SC18, SC19, and SC20) and three independent transgenic lines of tetraploid emmer wheat (RC12, RC26, and RC29) were tested for pathogen tolerance. The 1st, 2nd, and 3rd leaves were used for the analysis, and the results are presented in Figure 5 and Appendix A. Transgenic plants SC15 and SC20 were the most susceptible to *B. cinerea* among the tested hexaploid lines, demonstrating significantly lower tolerance compared to non-transgenic controls. The tolerance of SC19 plants did not differ significantly from that of non-transgenic control plants (Figure 5a,b). As mentioned above, even though there is no statistical significance in the difference in wounding-induced jasmonate levels between studied transgenic lines of hexaploid wheat and non-transgenic control due to the large variations between individual Sar-60 plants, the most susceptible to the pathogen transgenic lines SC15 and SC20, on average, showed elevated wounding-induced levels of JA and JA-Ile. 

Similar results were obtained when the tetraploid emmer wheat plants were tested. All transgenic lines, including RC12 and RC29, in which the wounding-induced level of jasmonates is significantly higher than in non-transgenic control plants, are more susceptible to the studied pathogen (Figure 5c–e and Appendix A). The measurements of the necrotic and chlorotic zone sizes showed that the difference in the size of the chlorotic zone between the studied genotypes was more pronounced, while the difference in the size of the necrotic zone was smaller and, in some cases, not statistically significant (Figure 5d,e and Appendix A).

### 2.5. Effect of the Exogenously Applied MeJA on Disease Development and B. cinerea Growth on an Agar Plate

We have studied the effect of exogenous methyl jasmonate treatment on infection development. First, we checked whether methyl jasmonate could directly affect the growth of the fungal pathogen. The results of the analysis showed a very moderate direct effect of methyl jasmonate added to the growth medium on the *B. cinerea* growth on potato-dextrose agar, and only very high concentrations of MeJA (10 mM) had a strong inhibitory effect on the pathogen growth (Figure 6a,b). 

Then the effect of various concentrations of methyl jasmonate sprayed on the Sar-60 and Runo leaves before infection was studied (Figure 7 and Appendix A). None of the concentrations of methyl jasmonate exogenously applied to wheat leaves resulted in an improvement in tolerance to the pathogen. In contrast, treatment with MeJA concentrations higher than 10 μM resulted in a significant decrease in pathogen tolerance. A particularly noticeable effect of exogenous treatment with methyl jasmonate was the formation of the chlorotic zone. Interestingly, there was no clearly defined necrotic zone on jasmonate-treated leaves; the area of brownish necrotic lesions was very small and did not extend beyond the site of infection, that is, the area of droplets with pathogen spores applied to the leaf. At the same time, it is obvious that the observed leaf yellowing is related to the infection process since the development of the chlorotic zone starts at the site of infection. This effect was clearly seen on both hexaploid and tetraploid wheat leaves, on the 1st, 2nd, and 3rd leaves that differed in age (Figure 7 and Appendix A). 

We also tested whether exogenously applied jasmonates affect leaf yellowing in the absence of infection. Figure 8 shows the effect of MeJA on the yellowing of the fourth leaf cut from plants at the 4-leaf stage. The youngest leaves have been used in this test to minimize the contribution of natural senescence. It is obvious that jasmonates do cause leaf yellowing, and this effect depends on the applied MeJA concentration (Figure 8). In this case, the chlorotic changes are accompanied by a rapid degradation of photosynthetic pigments. A similar and even stronger effect was observed on older leaves (Appendix A).

Finally, we conducted experiments confirming that the observed effects are not attributable to the experimental conditions or the used pathogen culture. To do this, we employed *Arabidopsis thaliana* plants, wild-type Columbia-0 ecotypes with intact jasmonate biosynthesis pathways, and *aos ko* mutant plants lacking jasmonates due to the mutation in *ALLENE OXIDE SYNTHASE*, the key gene of the jasmonate biosynthesis pathway. The *aos ko* plants exhibited significantly greater sensitivity to *B. cinerea*, and exogenous MeJA treatment significantly improved the mutant plants’ tolerance to the pathogen (Figure 9). Therefore, in accordance with numerous previously published data, in our experiments, jasmonates, both endogenously produced and exogenously applied, protected Arabidopsis plants from *B. cinerea* infection. Importantly, the MeJA treatment caused leaf yellowing neither in the wild type nor in the *aos ko* mutant.

## 3. Discussion

Plants rely on complex regulatory mechanisms to tightly control hormone levels under optimal and unfavorable environmental conditions. Therefore, even though all the enzymes of the jasmonate biosynthesis pathway are well characterized, the generation of plants with altered hormone content is still a difficult task due to the complex regulatory network controlling the endogenous jasmonate levels, including positive and negative regulatory feedback loops and regulatory processes at the transcriptional and post-transcriptional levels [33,34]. The *OPR* gene is one of the promising targets for genetic manipulation. Overexpression of *OsOPR7*, coding for the peroxisome-localized enzyme, in rice was shown to mitigate salinity stress [35]. In a recent study [36], the authors show that the gene dosage or expression level of monocot-specific OPR subfamily III members affects root architecture and their ability to access water under limited water conditions. Previously, we generated transgenic hexaploid wheat plants overexpressing the *AtOPR3* gene and displaying improved abiotic stress tolerance [30,37]. We used these previously established plants and generated new tetraploid emmer wheat plants (Figure 1 and Appendix A) to investigate the effect of the *AtOPR3* gene overexpression on wheat tolerance to biotic challenges. To assess the effect of *AtOPR3* overexpression on the jasmonate profile, we assessed the basal and stress-induced levels of 12-OPDA, JA, and JA-Il. Further, we addressed the stress-induced alteration in phytohormone levels by mechanically wounding plants (Figure 2 and Figure 3). Even though the wounding did not precisely mimic the stress-associated responses to the infection by a fungal pathogen, this approach is recognized as the most appropriate for assessing the potential of the jasmonate system. The wounding activates jasmonate biosynthesis and signaling, giving access to a uniform stress treatment [38,39]. On the other hand, the analysis of jasmonate contents in infected plants is prone to misinterpretation because the increased levels of the protective hormone may simply reflect the higher degree of pathogen-induced damage.

Analysis of non-transgenic hexaploid and tetraploid plants revealed characteristic features of the wheat jasmonate profile (Figure 2). Expectedly, the basal level of JA was very low. However, the extremely low contents of 12-OPDA appeared to be rather unexpected. The basal levels of 12-OPDA in both Sar-60 and Runo were below 20 pmol/g f.w., while the levels of 12-OPDA in Arabidopsis leaves varied between 0.5 and 10 nmol/g f.w., according to the different sources [38,40,41], which was tens or hundreds-fold higher than the values obtained in wheat. Detected levels of 12-OPDA were also much lower than those previously obtained for another monocotyledonous crop, rice, using the same analytical equipment and metabolite extraction and analysis methods [35,42]. Wounding led to an increase in the levels of all quantified jasmonates in both wheat species studied, but the magnitude of changes in the hormone content was also much smaller than that described for Arabidopsis, in which the wounding led to an increase in the levels of JA by a hundredfold or more. Nevertheless, using mechanical damage, we identified transgenic lines of tetraploid wheat with statistically significant increases in JA and JA-Ile levels, suitable for the study of the role of jasmonates in wheat pathogen tolerance.

Although *B. cinerea* does not cause significant losses in wheat, it can be used as a convenient model pathogen to determine plant tolerance to necrotrophic pathogens [43]. Our tests using *B. cinerea* have shown that jasmonates produced endogenously and applied exogenously provoke the development of disease symptoms in wheat leaves, primarily manifested in the formation of yellow (chlorotic) zones (Figure 5 and Figure 7). In order to make sure that the observed effects are not attributable to the experimental conditions or used pathogen culture, we reproduced the results, confirming that in Arabidopsis, endogenously produced and exogenously applied jasmonates exert a strong protective effect against *B. cinerea* (Figure 9). Although the data obtained on tetraploid and hexaploid wheat contradict most studies showing the protective function of jasmonates in plant tolerance to necrotrophic pathogens [44,45], there are still several examples in the literature similar to those obtained in the present work. A striking example is the study demonstrating the hyper-resistance of the *A. thaliana* mutants in JA biosynthesis and signaling pathways (*opr3*, *coi1*, and *jar1*) to the necrotrophic pathogen *Fusarium graminearum* [46]. The aforementioned study demonstrates that salicylic acid (SA) application and activation of systemic acquired resistance enhanced plant tolerance, while the jasmonate pathway contributed to susceptibility. The results of the study suggest that the JA pathway promotes disease by attenuating the activation of SA signaling in fungus-infected plants. It is generally accepted that salicylic acid, a jasmonate antagonist, regulates tolerance to biotrophic pathogens by inducing programmed cell death, and furthermore, the necrotrophic fungus *B. cinerea* induces SA-dependent responses to promote host cell death and antagonize JA-Ile-mediated resistance [47,48,49,50,51,52]. Yet, there is evidence in the current literature indicating that the processes of plant infection and defense mechanisms are much more complex and diverse. 

It was also shown that the AtMYC2 transcription factor, which regulates diverse JA-dependent processes, represses plant tolerance to pathogens [53]. MYC2 interacts with key regulators of ethylene signaling, ETHYLENE INSENSITIVE 3 (EIN3) and EIN3-like1 (EIL1), to repress tolerance to *B. cinerea*. 

Another interesting example is the study of plants with a modified jasmonoyl-isoleucine (JA-Ile) catabolic pathway mediated by cytochrome P450 (CYP94) enzymes [54]. Using knockout and overexpressing Arabidopsis lines with modified expression of the *CYP94B3* and *CYP94C1* genes, the authors showed that the JA-Ile catabolic pathway is an important component of the fungus-induced jasmonate pathway. In *CYP94* mutant plants, JA-Ile overaccumulation due to impaired oxidation had negligible impacts on resistance. *CYP94*-overexpressing plants accumulating products of JA-Ile catabolism still maintained JA-Ile levels at near wild-type levels but displayed impaired defense induction and increased susceptibility to the infection, suggesting that JA-Ile signaling may become non-functional due to the presence of an excessive amount of the oxidized form of the hormone. Moreover, feeding JA-Ile to seedlings leads to strong induction of JA pathway genes, while such induction is reduced or abolished after feeding with the JA-Ile oxidation products, 12-OH-JA-Ile and 12-COOH-JA-Ile. 

The most interesting and little-studied issue is the production of jasmonates by pathogens themselves. The only well-studied microbial jasmonate remains coronatine from the biotrophic pathogenic bacterium *Pseudomonas syringae*, which, upon entering the host plant tissues, activates JA signaling and suppresses SA-based plant defense [55,56]. Jasmonates were found in several fungal species, such as *Magnaporthe oryzae*, *Lasiodiplodia theobromae*, *Fusarium oxysporum*, and *Gibberella fujikuroi* [57,58,59,60]. Particularly important is the fact that JA and its derivate, 12-hydroxyjasmonic acid, are secreted by the fungus into plant tissue to suppress the JA-dependent host immunity [57], to stop the vegetative phase, and to induce pathogenic development [60]. 

A possible role for 12-OPDA in the regulation of pathogen tolerance cannot be excluded. Recently, using tomato plants with a silenced *OPR3* gene, it was shown that 12-OPDA, a substrate for the OPR3 enzyme, plays an important role in protection against the necrotrophic pathogen [61]. In maize, 12-OPDA-mediated resistance to a phloem sap-sucking corn leaf aphid, *Rhopalosiphum maidis*, was demonstrated to be independent of the jasmonic acid pathway [62]. However, in our work, we did not observe a significant decrease in the level of 12-OPDA in either hexaploid or tetraploid transgenic wheat lines, and, therefore, we cannot assume that the observed decrease in the tolerance to the pathogen is due to a decrease in this metabolite. Our data are consistent with those obtained on rice; in contrast to expectation, the overexpression of *OsOPR7* in rice does not reduce the steady-state levels of 12-OPDA, but rather leads to an increase [35]. The question about the role of 12-OPDA in the regulation of tolerance to pathogens requires further study using plants in which the level of 12-OPDA is markedly altered.

It has been observed in our study (Figure 4) and in a number of other works, that two processes can be distinguished in the development of the *B. cinerea* infection: the formation of necrotic damages near the infection site and distal spreading of the disease in the form of chlorotic damages associated with the secretion of hydrolytic enzymes, toxic compounds [63,64], or metabolites such as oxalic acid [65]. According to our data, *AtOPR3* overexpression and exogenous treatment with methyl jasmonate stimulate the formation of the chlorotic zone on the infected leaf. There are several molecular mechanisms responsible for the induction of leaf chlorosis. Chlorosis accompanies the development of viral and biotrophic infections [66,67]. It can be caused by accelerated or natural senescence, nutrition deficiency [68], and other instances. Interestingly, chlorosis was observed in wheat as a result of interspecific hybridization, and in these wheat hybrids, the plants with mild chlorosis showed increased resistance to wheat blast and powdery mildew fungi [69]. Biotrophic pathogens often delay senescence to keep host cells alive, and resistance to biotrophs can be achieved by the induction of senescence-like processes, while necrotrophic pathogens promote senescence in the host, and the plant resistance strategy in this case is to prevent early senescence [70]. For hemibiotrophic pathogens, which many consider *Botrytis cinerea* to be, both patterns may apply. 

The role of jasmonates in accelerating leaf senescence is known [67,71,72]. Jasmonates can promote leaf yellowing through their antagonistic interaction with cytokinin and nullify the protective effect of cytokinin on chlorophyll [73,74,75]. In our experiments, exogenous jasmonate treatment combined with and without infection resulted in a rapid yellowing of leaves, and this effect was clearly manifested on very young leaves as well (Figure 8 and Appendix A). It remains to be investigated whether this yellowing is due to the induction of early senescence or to a disturbance in the stability of the photosynthetic apparatus.

## 4. Materials and Methods

### 4.1. Generation of Transgenic Emmer Wheat Plants

The transformation of emmer wheat (cv. ‘Runo’) was performed following the particle bombardment method described by us earlier [76]. Since the pBAR-GFP.UbiOPR3 vector encodes two selective marker genes, *GFP* and *BAR*, a dual selection approach (*GFP* fluorescence and herbicide resistance of transgenic cells) was applied to produce transgenic wheat plantlets. Selected putative transgenic plants of emmer wheat were established in a greenhouse, and after confirmation that they demonstrate *GFP* fluorescence in pollen, the primary T0 transgenic plants were analyzed for *AtOPR3* integration and expression by end-point RT-PCR. RNA isolation and reverse transcription were carried out as described previously [30]. The primers used for the analysis are listed in Appendix A. 

Segregation of transgenes after self-pollination of T0 transgenic plants was carried out based on *GFP* expression. T1 seeds from the 15 hemizygous T0 plants with a distinctive expression of *GFP* in pollen were planted in the greenhouse and allowed to set T2 seeds. To identify homozygous transgenic plants, the pollen from individual T1 plants was screened for *GFP* fluorescence, as described previously [77]. T3–T4 progeny obtained from T2 homozygous transgenic families were further used for qPCR assays and pathogen tolerance analysis. 

### 4.2. Expression Analysis

Total RNA for qRT-PCR was extracted from the frozen third young leaves of greenhouse-grown T4 homozygous plants using TRIzol RNA Isolation Reagent (Thermo Fisher Scientific, Waltham, MA, USA) following the manufacturer’s instructions. The first-strand cDNA was synthesized from 2 μg of total RNA using RevertAid Reverse Transcriptase (Thermo Scientific). The reactions were based on SYBR Green qPCR SuperMix (Thermo Fisher Scientific) run in a QuantStudio™ 5 Real-Time PCR Cycler (Thermo Fisher Scientific) machine. The wheat *TaWIN1* gene [78] was used as a housekeeping gene for normalization. At least three biological replicates were run for each qPCR. The primers used for gene expression assays are listed in Appendix A.

### 4.3. Analysis of Phytohormones

The content of jasmonates was determined in the intact and wounded third leaf of plants at the 4-leaf stage. For this, the plants were planted in 1 L pots (three plants per pot) filled with soil. The plants were grown in a greenhouse under standard conditions (fertilized with Hoagland’s solution) and subjected to the analysis of jasmonate response to wounding. For this, the third leaf was clamped with forceps along its entire length at 10-cm intervals. The leaf tissues were harvested 30 min after wounding in parallel to untreated controls. After harvesting, the leaf material was immediately milled in liquid nitrogen, with a mortar and pistil before grounding in a Mixer Mill MM 400 ball mill with 3-mm-diameter stainless steel balls (Retsch, Haan, Germany) at a vibration frequency of 30 Hz for 2 min.

For determination of leaf jasmonate contents, approximately 50 mg of leaf material were extracted with 500 μL of ice-cold methanol containing 0.1 ng/μL of each stable isotope-labeled internal standard (^2^H_6_-JA, ^2^H_2_-(−)-JA-Ile and ^2^H_5_–OPDA) with subsequent pre-cleaning with solid phase extraction (SPE) using strong cation exchange HR-XC material (Macherey & Nagel, Düren, Germany) in a 96-well format as described by Balcke and co-workers [79]. The absolute quantification relied on the stable isotope dilution approach and triple quadrupole mass spectrometry of the resulted eluates using an ACQUITY H-Class UPLC ultrahigh performance liquid chromatography system (Waters GmbH, Eschborn, Germany) coupled on line to a QTRAP 6500 (AB Sciex, Darmstadt, Germany) triple quadrupole-linear ion trap instrument operating in negative multiple reaction monitoring (MRM) mode under the settings described by Leonova and co-workers [80].

### 4.4. B. cinerea Tolerance Test

To estimate the tolerance of the studied plants to *B. cinerea,* we employed a detached leaf assay. Wheat plants were grown at greenhouse conditions (25 ± 2 °C during the day and 20 ± 2 °C at night under a 16-h photoperiod with additional lighting during the winter period to provide a light intensity of up to 150 µmol/m^2^s). Five plants were grown in a 3-L pot filled with soil until the 4-leaf stage (about 4 weeks old). The 1st, 2nd, and 3rd leaves were used for the bioassay. *Arabidopsis thaliana* plants were grown in a short-day photoperiod (8-h-light/16-h-dark) at 22 °C. Mature rosette leaves excised from 5-week-old plants were used in the assay. 

*B. cinerea* spore inoculum was prepared as described earlier [81,82,83]. *Arabidopsis* rosette leaves with petioles or 9-cm fragments of excised wheat leaves were placed in square Petri dishes on wet filter paper. Due to the more rapid aging of cut wheat leaves, in wheat experiments, the filter paper was soaked with kinetin solution (10 mg/L). Leaves were inoculated with 4-µL droplets of 2 × 10^6^ spores/mL in filtered organic peach juice diluted with distilled water (1:3). Plates were incubated at room temperature under ambient light conditions. Lesion size was measured from digital images of infected leaves using Image J, with scaled objects included in the images. For imaging of infected leaves, the digital dissecting microscope Andonstar ADSM301 (Andonstar, Shenzhen, China) and the light microscope Biomed 6 (Russia) were used.

For exogenous oxylipin treatment, a water solution of methyl jasmonate of a specified concentration was sprayed on detached leaves twice every 90 min before the inoculation of pathogen spores. To study the effect of MeJA on *B. cinerea* growth on agar plates, MeJA was added to the plates prior to plug inoculation.

### 4.5. Trypan Blue Staining

Infected leaf samples were fixed in a solution of 60% methanol, 30% chloroform, and 10% acetic acid and then rehydrated using solutions with decreasing concentrations of ethanol (100, 80, 70, and 50%). Rehydrated samples were stained with 0.05% trypan blue (Life Technologies, Eugene, OR, USA) in water overnight. Pictures were taken after de-staining.

## 5. Conclusions

The presented work demonstrates that jasmonate-mediated responses in polyploid wheat species may differ significantly from those described in dicotyledonous plants and adds data to the collection of information on the complexity of jasmonate-related regulation. The data obtained make us more cautious in applying the accepted model of jasmonate-mediated plant tolerance to necrotrophic pathogens to agricultural crops.

## Figures and Tables

**Figure 1 plants-12-02050-f001:**
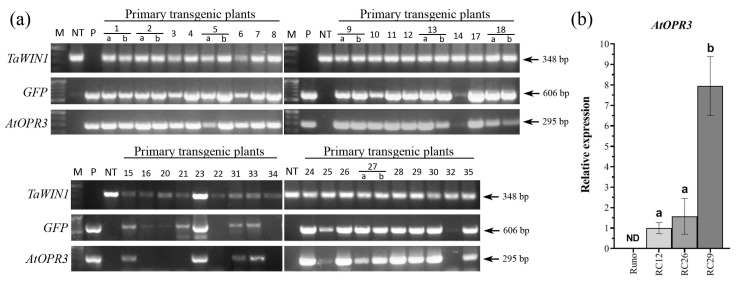
Molecular analysis of transgenic plants of emmer wheat “Runo”. (**a**) End-point RT-PCR analysis of the total RNA of T0 plants (at heading stage) for the expression of the reference gene *TaWIN1*, reporter gene *GFP,* and *AtOPR3*. Lane M, DNA ladder as a molecular weight marker; Lane P, plasmid DNA pBAR-GFP.UbiOPR3; Lane NT, non-transgenic wheat plant Runo; Lanes labeled 1–35 represent primary putative transgenic wheat plants established in a greenhouse; in a few cases, two plantlets were regenerated from one explant, and both plants (indicated as a or b) were independently analyzed for transgene insertion and expression; (**b**) relative expression levels of *AtOPR3* in leaves of three transgenic emmer wheat lines, T4 homozygous plants; data are means of four biological replicates ± SE; ND—not detected; different letters above the graphs indicate statistically significant differences (*p* ≤ 0.05) assessed by the one-way ANOVA test.

**Figure 2 plants-12-02050-f002:**
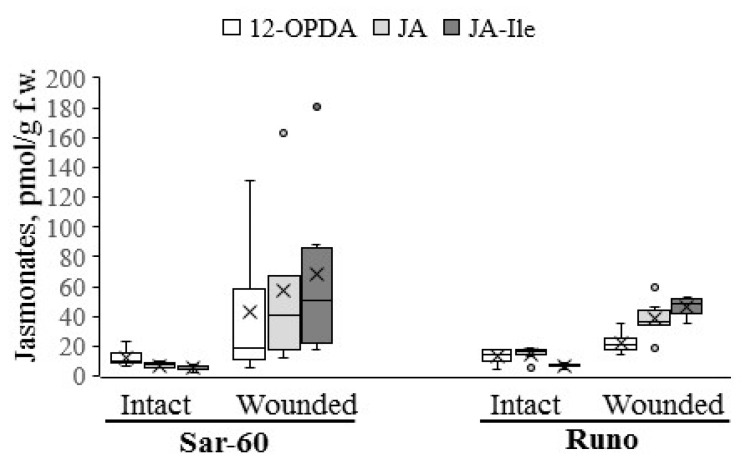
The content of individual jasmonates (12-OPDA—white bars; JA—light gray bars; JA-Ile—dark gray bars) in intact and wounded leaf tissues of hexaploid (Sar-60) and tetraploid (Runo) wheat plants. Each box represents data from six biological replicates with whiskers extended to the extreme data points, where the midline is the median, the cross is the mean, and dots are outliers.

**Figure 3 plants-12-02050-f003:**
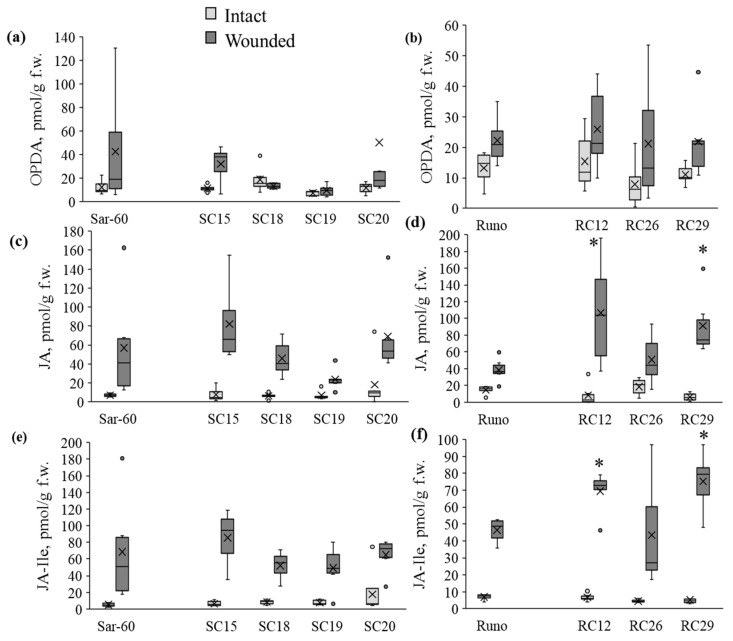
The content of jasmonates ((**a**,**b**), 12-OPDA; (**c**,**d**), JA; (**e**,**f**), JA-Ile) in intact (light grey bars) and wounded (dark grey bars) leaf tissues from transgenic lines of hexaploid Sar-60 (**a**,**c**,**e**) and tetraploid Runo (**b**,**d**,**f**) wheat. Each box represents data from six biological replicates with whiskers extended to the extreme data points; the midline is the median, the cross is the mean, and dots are outliers. Stars indicate a statistically significant difference from the non-transgenic control at *p* ≤ 0.05, as assessed by Student’s t-test.

**Figure 4 plants-12-02050-f004:**
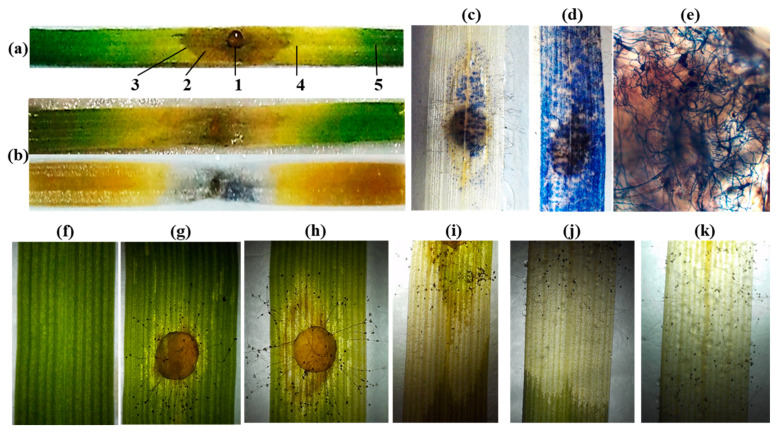
*B. cinerea* infection symptoms on a detached wheat leaf. (**a**) Representative image of an infected leaf showing an applied drop with fungal spores (1), the brownish necrotic area (2) separated by a clear boundary (3) from the chlorotic area (4), and the green tissue with no visible symptoms of the disease (5); (**b**) trypan blue staining of an infected leaf, before (top) and after (bottom) staining; high resolution macroscopic (**c**,**d**) and microscopic (**e**) images of trypan blue stained leaves; macroscopic images of non-infected (**f**) and infected leaves at different stages of *B. cinerea* infection development (**g**–**k**).

**Figure 5 plants-12-02050-f005:**
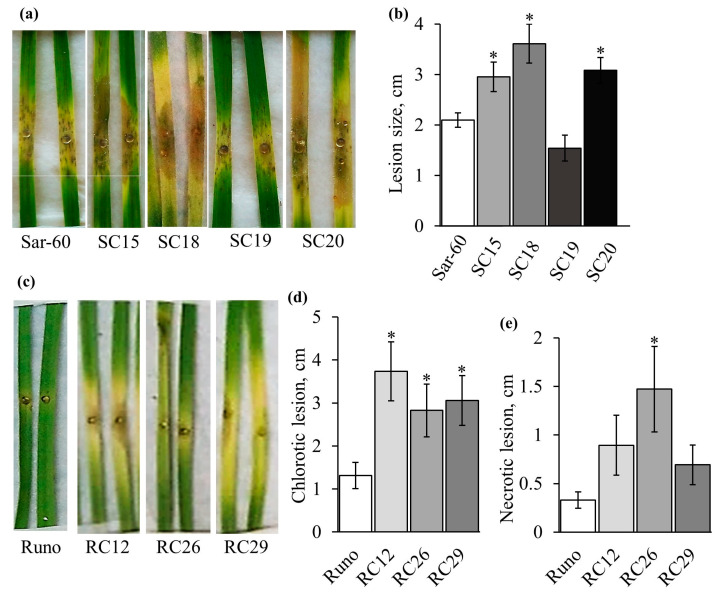
Tolerance of transgenic wheat plants to *B. cinerea*. Representative images of *B. cinerea*-infected leaves of hexaploid (**a**) and tetraploid (**c**) wheat on the 5th day after infection. Length of the lesions ((**b**) for Sar-60 genotypes, (**d**,**e**) for Runo genotypes). The data are averages of 3–5 biological replicas consisting of 4–12 plants ± SE. Stars indicate a statistically significant difference from the non-transgenic control at *p* ≤ 0.05, as assessed by Student’s t test.

**Figure 6 plants-12-02050-f006:**
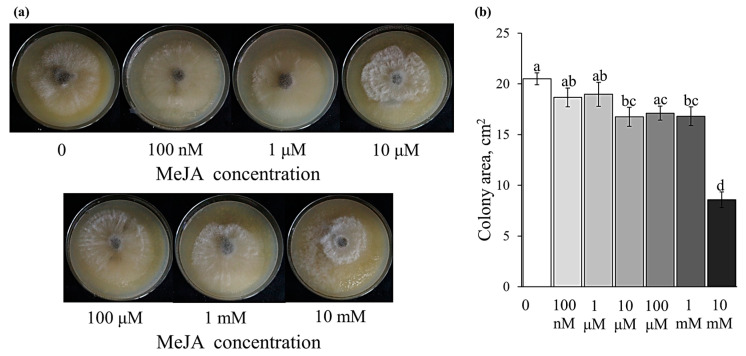
Effect of MeJA on *B. cinerea* growth on an agar plate. (**a**) Images of *B. cinerea* culture growing on Petri dishes with different MeJA concentrations on the third day after agar plug inoculation; (**b**) *B. cinerea* colony (developed aerial mycelium) size presented as the average of two replicates with a standard deviation; different letters above the bars indicate a statistically significant difference between the treatments at *p* ≤ 0.05 assessed by the one-way ANOVA test.

**Figure 7 plants-12-02050-f007:**
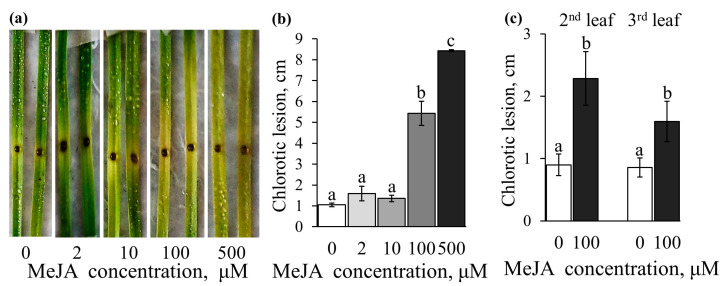
Effect of exogenously applied MeJA on *B. cinerea* infection development on detached leaves of hexaploid (**a**,**b**) and tetraploid (**c**) wheat. (**a**) Representative images of *B. cinerea*-infected Sar-60 leaves on the 3rd day after infection; (**b**) length of the lesion zone. Data are averages of 10–22 replicates ± SE; different letters above the bars indicate a statistically significant difference between the treatments at *p* ≤ 0.05, as assessed by the one-way ANOVA test.

**Figure 8 plants-12-02050-f008:**
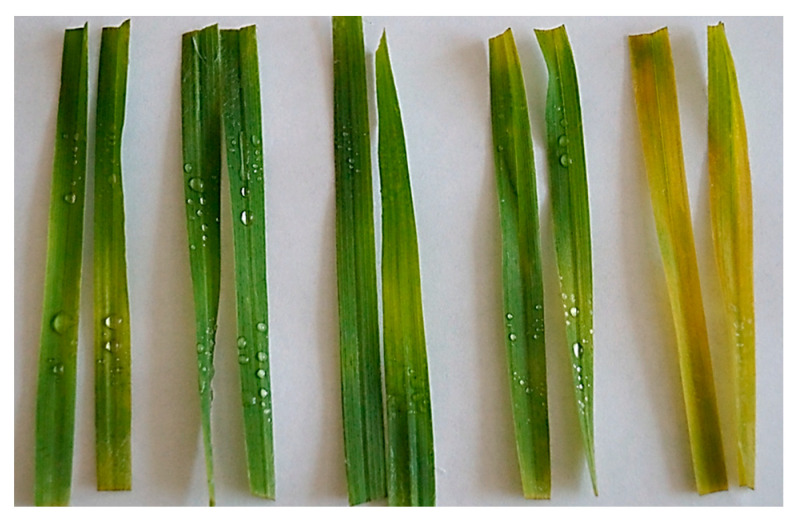
Effect of MeJA treatment on detached Sar-60 leaves yellowing. The chlorophyll content of the presented leaves is shown below the photograph. 0, 2, 10, 100, 500, MeJA concentration, μM. 1.32, 0.78, 0.77, 0.4, n.d. Chlorophyll concentration, mg/g f.w.

**Figure 9 plants-12-02050-f009:**
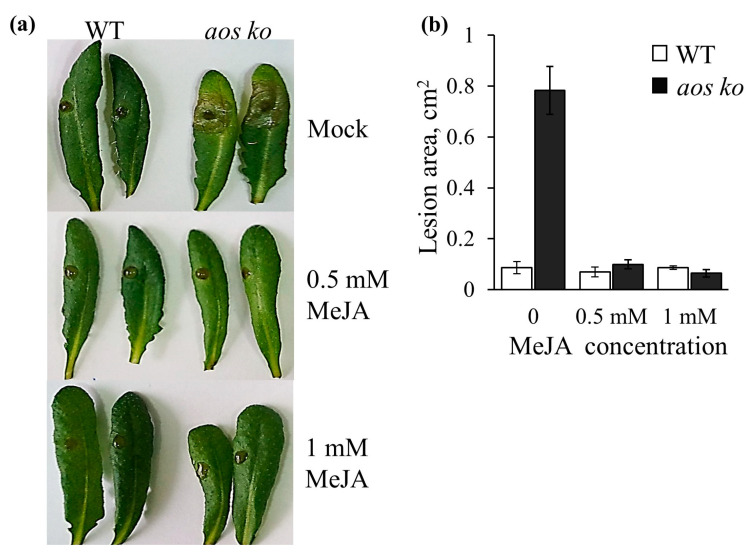
Effect of endogenously produced and exogenously applied jasmonates on *B. cinerea* infection development on detached *Arabidopsis thaliana* leaves, wild-type Columbia-0 ecotype (WT), and mutant lines depleted of the functional *ALLENE OXIDE SYNTHASE* gene (*aos ko*). (**a**) Representative images of infected leaves; (**b**) lesion size on the leaves on the third day after the pathogen spore inoculation. The data are averages of nine measurements ± SE.

## Data Availability

The data presented here are available upon request from the corresponding authors.

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
