# Peer review of "12-Oxophytodienoate Reductase Overexpression Compromises Tolerance to Botrytis cinerea in Hexaploid and Tetraploid Wheat"

_plants, 2023, doi:10.3390/plants12102050_

Round 1

Reviewer 1 Report

In this manuscript Degtyaryov et al report on overexpression of AtOPR3 in wheat in relation to susceptibility to Botrytis. Overall, the manuscript is well-written and sufficiently documented. I do have some questions and remarks:

Major

Very recently a paper was published by Gabay et al on OPR3 genes in Wheat which should be discussed

Figure 1B        As AtOPR3 is absent in Wheat, plotting relative expression compared to the control does not make any sense. RC12 could be set to 1. Alternatives here are Northern blot for example. As the authors have some unexpected results with the lines, I wonder if the AtOPR3 has any nucleotide homology with TaOPR-genes. Did they check expression of those? Maybe there is co-supression.

For the analysis of JA content, also in the result explain briefly what kind of material you use. As these are greenhouse-grown, this might explain the variation.

The authors in the discussion themselves highlight the very low amounts of Jas detected. Did they include a technical control such as a Arabidopsis sample to make sure the detection and calculations are correct? In their previous paper (Pigolev et al) they plot Jas as ng/g FW as many other authors do. Why did they change? Is there in the literature examples of JA content in wheat?

Minor

Figure 1S         What is the s in sGFP? Also list terminators for GFP and BAR. AmpR in legend, but not figure.

Figure 5           Statistically significant using which test?

Discussion       L384 OPR3 stimulates formation of chlorotic zone > only if challenged, OE itself does not cause this I assume

Author Response

Dear Reviewer,

We thank you for the in-depth analysis of our article and for your valuable suggestions. We made all the required changes, and it helped us improve the article. Below are the responses to each comment.

Very recently a paper was published by Gabay et al on OPR3 genes in Wheat which should be discussed

Following your suggestion, we have mentioned the work by Gabay et al in the discussion (L312-314 of the revised version of the manuscript), but we did not discuss the presented in this article data extensively, because the studied by Gabay and co-authors OPR genes belong to the Subfamily III, while wheat homologs of AtOPR3 with proven role in jasmonates biosynthesis belong to the Subfamily II (Mou, Y.; Liu, Y.; Tian, S.; Guo, Q.; Wang, C.; Wen, S. Genome-Wide Identification and Characterization of the OPR Gene Family in Wheat (Triticum aestivum L.). Int. J. Mol. Sci. 201920, 1914). Moreover, the authors have mentioned: “We show here that the understudied genes from the monocot-specific OPRIII subfamily encode cytoplasmic and nuclear 12-OXOPHYTODIENOATE REDUCTASE enzymes that regulate a critical step in the synthesis of JA-Ile”.  All previously described OPR enzymes, involved in JA biosynthesis, including AtOPR3, are localized in peroxisomes.

Figure 1B        As AtOPR3 is absent in Wheat, plotting relative expression compared to the control does not make any sense. RC12 could be set to 1. Alternatives here are Northern blot for example. As the authors have some unexpected results with the lines, I wonder if the AtOPR3 has any nucleotide homology with TaOPR-genes. Did they check expression of those? Maybe there is co-supression.

Thank you for pointing to this mistake. It is a result of the technical error in the data presentation. In the revised manuscript, we are presenting the corrected figure. The primers we used are very specific to AtOPR3 and do not amplify other sequences. The presence of the single PCR product has been confirmed by electrophoresis in agar gel and PCR product melting curve analysis.

Concerning the question about the homology between AtOPR3 and TaOPR genes: there are 48 putative members in wheat OPR family, and among them 4 OPRII subfamily members, which share moderate homology of the nucleotide sequence with AtOPR3. In our previous work (Pigolev et al., 2018), we have analyzed the expression of the endogenous wheat OPR genes using primers specific to TaOPRII, and found that the expression of this gene/genes is not suppressed in the transgenic lines overexpressing AtOPR3.

For the analysis of JA content, also in the result explain briefly what kind of material you use. As these are greenhouse-grown, this might explain the variation.

The corresponding text is added to the Results sections (Lines 127-130)

The authors in the discussion themselves highlight the very low amounts of Jas detected. Did they include a technical control such as a Arabidopsis sample to make sure the detection and calculations are correct? In their previous paper (Pigolev et al) they plot Jas as ng/g FW as many other authors do. Why did they change? Is there in the literature examples of JA content in wheat?

We did use Arabidopsis leaves as a control and observed significantly higher levels of jasmonic acid and a strong induction of the hormone level by wounding. Moreover, as we mention in the modified text (Lines 341-343), “Detected levels of 12-OPDA were also much lower than those previously obtained for another monocotyledonous crop, rice, using the same analytical equipment and metabolite extraction and analysis methods [35,42]”. According to our previous publication (Pigolev et al 2018), in intact hexaploid wheat leaf the JA level is 3 ng/g f.w., which is equal 14 pmol/ g f.w.  This is lower than the commonly reported jasmonate content of Arabidopsis. In the work (Pigolev et al 2018), we did not measure 12-OPDA level and stress-induced hormone levels. In the work presented, we assure that the content of 12-OPDA and the degree of induction of jasmonate levels by wounding in wheat are low.

Minor

Figure 1S         What is the s in sGFP? Also list terminators for GFP and BAR. AmpR in legend, but not figure.

Corrected.

Figure 5           Statistically significant using which test?

The information is added to the Figure legends.

 Discussion       L384 OPR3 stimulates formation of chlorotic zone > only if challenged, OE itself does not cause this I assume

This important clarification is added to the text.

Reviewer 2 Report

The paper titled "12-Oxophytodienoate Reductase Overexpression Compromises Tolerance to Botrytis Cinerea in Hexaploid and Tetraploid Wheat" demonstrates that jasmonate-mediated responses in polyploid wheat species may significantly differ from those described in dicotyledonous plants. The study adds valuable data to the existing information on the complexity of jasmonate-related regulation. The methods were well-described, and the results were presented clearly. However, there are some areas of concern, particularly in the conclusion section, that require addressing.

(1) The jasmonate family of growth regulators includes the isoleucine (Ile) conjugate of jasmonic acid (JA-Ile) and its biosynthetic precursor, 12-oxophytodienoic acid (OPDA), as signaling molecules. Loredana Scalschi et al. (2014, Plant J, 81:304-315) provide evidence that OPDA by itself plays a major role in the basal defense of tomato plants against necrotrophic pathogens. Suresh Varsani et al. (2019, Plant Physiology, Volume 179, Issue 4, April 2019, Pages 1402–1415) reported that OPDA-mediated resistance to CLA is independent of the jasmonic acid pathway. Overexpression of 12-Oxophytodienoate reductase from Arabidopsis thaliana may decrease OPDA content, which could contribute to a compromise in tolerance to Botrytis Cinerea in hexaploid and tetraploid plants. Please consider these hypotheses and provide more evidence of the effects of decreasing OPDA on tolerance to B. Cinerea in hexaploid and tetraploid plants.

(2) Please revise the abstract according to the submission instructions of the journal, avoiding a discussion-style approach.

(3) Please distinguish between the yellowing (chlorophyll degradation) caused by jasmonic acid and the infection size caused by the development of B. cinerea infection in Figure 7. The brownish necrotic area or size of trypan blue stained leaves may be more suitable.

Author Response

Dear Reviewer,

Thank you very much for the thorough review of the manuscript. We appreciate all your comments

and suggestions. We have introduced all required changes. Below are the itemized responses.

(1) The jasmonate family of growth regulators includes the isoleucine (Ile) conjugate of jasmonic acid (JA-Ile) and its biosynthetic precursor, 12-oxophytodienoic acid (OPDA), as signaling molecules. Loredana Scalschi et al. (2014, Plant J, 81:304-315) provide evidence that OPDA by itself plays a major role in the basal defense of tomato plants against necrotrophic pathogens. Suresh Varsani et al. (2019, Plant Physiology, Volume 179, Issue 4, April 2019, Pages 1402–1415) reported that OPDA-mediated resistance to CLA is independent of the jasmonic acid pathway. Overexpression of 12-Oxophytodienoate reductase from Arabidopsis thaliana may decrease OPDA content, which could contribute to a compromise in tolerance to Botrytis Cinerea in hexaploid and tetraploid plants. Please consider these hypotheses and provide more evidence of the effects of decreasing OPDA on tolerance to B. Cinerea in hexaploid and tetraploid plants.

Thank you very much for this suggestion. We have introduced additional paragraph to the discussion section (Lines 399-411).

(2) Please revise the abstract according to the submission instructions of the journal, avoiding a discussion-style approach.

We have changed the abstract and removed discussion-style phrases.

(3) Please distinguish between the yellowing (chlorophyll degradation) caused by jasmonic acid and the infection size caused by the development of B. cinerea infection in Figure 7. The brownish necrotic area or size of trypan blue stained leaves may be more suitable.

Thank you for this note. We have introduced the changes to the Figure 7 and the information that “the area of brownish necrotic lesions was very small and did not extend beyond the site of infection, that is the area of droplet with pathogen spores applied to the leaf” (Lines 256-258).

Round 2

Reviewer 2 Report

There is no more concern about the MS.